# Catalytic Conversion of Glycerol into Hydrogen and Value-Added Chemicals: Recent Research Advances

Yulin Hu [1], Quan He [2] and Chunbao Xu [3,*]

1 Faculty of Sustainable Design Engineering, University of Prince Edward Island, 550 University Ave, Charlottetown, PE C1A 4P3, Canada; yulinhu@upei.ca
2 Department of Engineering, Faculty of Agriculture, Dalhousie University, Truro, NS B2N 5E3, Canada; quan.he@dal.ca
3 Department of Chemical and Biochemical Engineering, Western University, 1151 Richmond St., London, ON N6A 3K7, Canada
* Correspondence: cxu6@uwo.ca

**Abstract:** In recent decades, the use of biomass as alternative resources to produce renewable and sustainable biofuels such as biodiesel has gained attention given the situation of the progressive exhaustion of easily accessible fossil fuels, increasing environmental concerns, and a dramatically growing global population. The conventional transesterification of edible, nonedible, or waste cooking oils to produce biodiesel is always accompanied by the formation of glycerol as the by-product. Undeniably, it is essential to economically use this by-product to produce a range of valuable fuels and chemicals to ensure the sustainability of the transesterification process. Therefore, recently, glycerol has been used as a feedstock for the production of value-added $H_2$ and chemicals. In this review, the recent advances in the catalytic conversion of glycerol to $H_2$ and high-value chemicals are thoroughly discussed. Specifically, the activity, stability, and recyclability of the catalysts used in the steam reforming of glycerol for $H_2$ production are covered. In addition, the behavior and performance of heterogeneous catalysts in terms of the roles of active metal and support toward the formation of acrolein, lactic acid, 1,3-propanediol, and 1,2-propanediol from glycerol are reviewed. Recommendations for future research and main conclusions are provided. Overall, this review offers guidance and directions for the sufficient and economical utilization of glycerol to generate fuels and high value chemicals, which will ultimately benefit industry, environment, and economy.

**Keywords:** glycerol; catalysts; $H_2$; chemicals; sustainability

## 1. Introduction

Until now, considerable effort has been dedicated toward developing renewable resources to completely or partially replace with fossil fuels, including wind, solar, geothermal, nuclear, tidal power, and biomass, among which biomass is regarded as the best energy precursor, especially given the introduction of the concept of biofuels [1]. Biofuels can be categorized into solid (e.g., pellets, briquettes, and biochar), gas (e.g., biohydrogen, biogas, and biomethane), and liquid (e.g., bioethanol, biobutanol, bio-oil, and biodiesel) fuels, and they are readily distributed as energy carriers within the existing infrastructure. For example, biodiesel obtained from animal fats, vegetable oils, or waste cooking oils via transesterification (Figure 1) can be directly used in diesel engines without modification or blending with diesel fuel. Owing to the changes in the energy and environmental landscapes, stringent environmental regulations have been imposed by governments, e.g., B-5, which is composed of 5% of biodiesel and 95% of diesel fuel, is primarily used in Canada to reduce greenhouse gas (GHG) emissions and other toxic gas emissions caused by burning fossil fuels, which has boosted biodiesel production in recent decades [2]. Clearly, this blooming production of biodiesel results in a glut of glycerol and a huge quantity of generated waste. Approximately, for each 1000 kg biodiesel produced, 110 kg of crude glycerol

is generated as the low-value by-product [3]. It was estimated that the global glycerol production reached approx. 4.2 million tons in 2020; however, the demand for glycerol was lower than 3.5 million tons, causing a large quantity of crude glycerol to be considered as a waste [4]. Consequently, efficient valorization technologies must be developed to convert glycerol into useful products rather than disposing them as waste. Because of its high functionalization, glycerol can be valorized into a wide range of products via multiple conversion routes, as discussed by Katryniok et al. [5]. Table 1 summarizes the high-value products, including fuels and chemicals, that can be feasibly produced from glycerol. In addition to $H_2$ and chemicals, glycerol can be applied as feed to react with free fatty acids to form glycerides by glycerolysis (also called glycerol esterification), and the resulting glyceride can be further treated to produce biodiesel via alkaline transesterification. Nevertheless, owing to the use of high-cost metallic catalysts and higher temperature (up to ~200 °C), glycerolysis is not a technology commonly used in the biodiesel industry, rather being widely employed in the cosmetic, pharmaceutical, and food industries to synthesize surfactants and emulsifiers [6]. The associated underlying mechanism, major reaction conditions (temperature, reactor configuration, molar ratio of glycerol and free fatty acid, type of free fatty acid, catalyst type, and glycerol purity), and technical challenges have been recently reviewed by Mamtani et al. [7] and Abomohra et al. [8].

**Figure 1.** The chemical reaction scheme for transesterification of triglyceride fatty acid methyl ester (FAME) and glycerol as the by-product.

**Table 1.** Summary of value-added products obtained from glycerol by catalytic routes.

| Reaction | Catalyst | Product | Reference |
|---|---|---|---|
| Dehydration | MoP | Acrolein | [9] |
| | Mo-V/ZSM-5 | Acrylic acid | [10] |
| | Au-Pt/$Al_2O_3$ | Lactic acid | [11] |
| | Au-Pt/$Al_2O_3$ | Glyceric acid | [11] |
| Oxidation | Cu-Mg | Glycolic acid | [12] |
| | Cu/$Al_2O_3$ | Oxalic acid | [13] |
| | $CoO_x$ | Dihydroxyacetone | [14] |
| | $WO_3$/$TiO_2$ | Glyceraldehyde | [15] |
| | Ru-Cu/CNT | 1,2-Propanediol | [16] |
| Hydrogenolysis | Pt/W-MCFs | 1,3-Propanediol | [17] |
| | Ni/$WO_3$-$TiO_2$; Ni/$WO_3$-$ZrO_2$ | 1-Propanol | [18] |
| Steam reforming | Pt/CCO | $H_2$-rich syngas | [19] |
| Esterification | $ZrO_2$/MCM-41 | Glycerides | [20] |
| Etherification | Zeolites; heteropolyacids; TSA/$SiO_2$; TPA/$SiO_2$; TSA/MCM-41; TSA/SBA-15; TPA/MCM-41; TSA/SBA-15; ion exchange resins | Methyl tert-butyl ether | [21] |
| Polymerization | | Polymers | [22] |

To enhance the efficiency of the valorization routes, the role of catalysts is vital, as evidenced by a surge in the number of related publications. Unlike transesterification, where chemical routes are clear with well-established catalysts, glycerol conversion routes are broad and versatile; therefore, it is challenging to cover a vast field of the relevant

research. Thus, in this review article, recent advances in glycerol valorization into $H_2$ by steam reforming and various chemicals, including acrolein by dehydration, lactic acid by oxidation, and 1,2-propanediol and 1,3-propanediol by selective hydrogenolysis, are discussed. Considering the chemical structure of the glycerol molecule, the activation and reactivity of C-C, C-O, C-H, and O-H bonds play an important role in the selection of reaction conditions and catalytic performance. For example, for C-C and C-O cleavage, bifunctional catalysts consisting of active metals and support are favorable for the reaction, among which a metallic complex or noble catalyst such as Pt, Pd, and Rh over acid catalyst support such as zeolites and activated carbon is the most common in the glycerol conversion. Conversely, in the case of C-H and O-H cleavage, it is critical to select a suitable metallic active center, among which noble metals and transition metals such as Cu, Ni, and Co are mainly utilized [23].

To date, most published review articles are restricted to one specific glycerol valorization technique, e.g., 1,3-propanediol by Wang et al. [24] and Eokum et al. [25]; lactic acid by Arcanjo et al. [26]; fuel additives by Smirnov et al. [27], Cornejo et al. [28], and Nanda et al. [29]; acrolein by Galadima and Muraza [30]; and $H_2$ and syngas by He et al. [4], Lin [31], and Macedo et al. [32].On the contrary, in this review, we discuss the latest developments and advances in heterogeneous catalysts and reactor configurations for some of the most common conversion pathways including steam reforming to $H_2$, dehydration to acrolein, oxidation to lactic acid, and selective hydrogenolysis to 1,3-propanediol and 1,2-propanediol, followed by a discussion on the directions for future research and major conclusions.

## 2. Utilization of Glycerol as Feedstock to Produce $H_2$

$H_2$ can act as an alternative energy carrier to replace fossil fuels and it can be produced via steam reforming of fossil fuels or biomass and water electrolysis. In industry, steam reforming of methane (SRM) for $H_2$ production is the dominant technology, which represents around 48% of the total $H_2$ production in the world [33]. Recent investigations regarding the glycerol conversion for producing $H_2$ via steam reforming over a transition metal or noble metal-based catalyst are summarized in Tables 2 and 3, respectively. The underlying reaction mechanism for glycerol steam reforming was illustrated by Sahraei et al. [34], as depicted in Figure 2.

**Table 2.** Recent investigations on glycerol conversion for producing $H_2$ via steam reforming over transition metal-based catalysts.

| Catalyst | Conditions | Max. Glycerol Conversion (%) | Max. $H_2$ Selectivity (%) | Reference |
|---|---|---|---|---|
| Ni/upgraded slag oxide | 480 °C and 580 °C; water/glycerol molar ratio of 9 | / | / | [35] |
| Ni/coal fly ash | 630 °C; water/glycerol molar ratio of 9; WHSV of 6.47 $h^{-1}$ | 93 | 77 | [36] |
| Ni-MgO/attapulgite | 400–800 °C; steam/carbon molar ratio of 3; WHSV of 1 $h^{-1}$ | 95 | 82 | [37] |
| Single Ni/SiO$_2$; Single CuSiO$_2$; Dual Ni/SiO$_2$-CuSiO$_2$ | 300–600 °C; Feed rate of 0.12 mL/min; LHSV of 7.6 $h^{-1}$ | 100 | 80 | [38] |
| Ni/MCM-$_{41}$; Ni/SBA-$_{15}$; Ni/CeO$_2$-MCM-$_{41}$; Ni/CeO$_2$-SBA-$_{15}$ | 650 °C; steam/carbon molar ratio of 2 | 99 | 92 | [39] |

**Table 2.** *Cont.*

| Catalyst | Conditions | Max. Glycerol Conversion (%) | Max. H$_2$ Selectivity (%) | Reference |
|---|---|---|---|---|
| Co/MgO; Cu-Co/MgO; Co/MgO-Al$_2$O$_3$ | 500–650 °C; WHSV of 2.88 h$^{-1}$ | 100 | 75 | [40] |
| Ni/La$_2$O$_3$-Al$_2$O$_3$; Ni/CeO$_2$-Al$_2$O$_3$; Ni/MgO-Al$_2$O$_3$; Ni/CeO$_2$-ZrO$_2$; | 500 °C; glycerol loading of 20 wt %; feed rate of 0.5 mL/min; catalyst loading of 5 wt % | 87 | 67 | [41] |
| Ni/amZr; Ni/Zr703; Ni/Zr873; Ni/9YSZ | 550 °C; glycerol loading of 20 wt %; | / | / | [42] |
| Co/MgO-Al$_2$O$_3$ | 500 °C; GHSV of 200,000 h$^{-1}$; glycerol loading of 20 vol % | 65 | 37 | [43] |
| Ni/SBA-15; Ni/La$_2$O$_3$-SBA15; Ni/La$_2$O$_3$-CeO$_2$-SBA15; Ni/La$_2$O$_3$-CeO$_2$-KIT-6 | 650 °C; LHSV of 2.8–11.3 h$^{-1}$ | / | 62 | [44] |
| Ni/Mg-Al | 400–700 °C; water/glycerol molar ratio of 9; feed rate of 0.025 mL/min | 30 | / | [45] |
| Ni-Co/CNT | 525 °C; glycerol loading of 10 wt %; feed rate of 6.0 mL/min | 96 | 94 | [46] |

**Table 3.** Recent investigations on glycerol conversion for producing H$_2$ via steam reforming over noble-based catalysts.

| Catalyst | Conditions | Max. Glycerol Conversion (%) | Max. H$_2$ Selectivity (%) | Reference |
|---|---|---|---|---|
| Pt-Ni/MgAl$_2$O$_4$ | 700–850 °C; Water/glycerol molar ratio of 12 | 100 | / | [47] |
| Rh/CeO$_2$-Al$_2$O$_3$; Rh/MgO-Al$_2$O$_3$; Rh/La$_2$O$_3$-Al$_2$O$_3$ | 400–750 °C; glycerol loading of 20 vol %; feed rate of 0.12 NmL/min; WHSV of 50,000 NmL/g· h | 90 | 78 | [48] |
| Pd/CeO$_2$-Al$_2$O$_3$; Pt/CeO$_2$-Al$_2$O$_3$ | 400–750 °C; water/glycerol molar ratio of 20 | 95 | 94 | [49] |
| Pt/SiO$_2$-C; Pt/SiO$_2$; Pt/C | 450 °C; glycerol loading of 10–50 wt %; steam/carbon molar ratio of 1.6–15; WHSV of 2.9–25.7 h$^{-1}$ | 100 | 75 | [50] |

**Table 3.** *Cont.*

| Catalyst | Conditions | Max. Glycerol Conversion (%) | Max. $H_2$ Selectivity (%) | Reference |
|---|---|---|---|---|
| Rh/MgO-Al$_2$O$_3$; Ru/MgO-Al$_2$O$_3$; Pt/MgO-Al$_2$O$_3$ | 300–600 °C; water/glycerol molar ratio of 9; GHSV of 35,000 mL/g· h | 100 | 100 | [51] |
| Pt-Sn/C | 350–400 °C; feed rate of 0.05 mL/min; glycerol loading of 10–30 wt % | 100 | 45 | [52] |
| Pt/VO$_x$-Al$_2$O$_3$ | 400 °C; feed rate of 1.9 mL/h; glycerol loading of 3.3 mol/h | / | / | [53] |
| Pt/SiO$_2$ | 300–400 °C; feed rate of 3–7 mL/h; WHSV of 47.25–110.25 h$^{-1}$; steam/carbon molar ratio of 3 | 97 | 97 | [54] |
| Ru/upgraded slag oxide metallurgical waste; Rh/upgraded slag oxide metallurgical waste | 630 °C; Water/glycerol ratio of 9; feed rate of 0.05 mL/min; GHSV of 10,966 cm$^3$/g$_{cat}$$^{-1}$ h$^{-1}$ | 100 | 78 | [55] |
| Rh/Al$_2$O$_3$ | 400 °C | 99 | / | [56] |
| Ru-Ni/CeO$_2$-Al$_2$O$_3$ | 550–800 °C; WHSV of h$^{-1}$ | / | 89 | [57] |
| Rh/MgAl$_2$O$_4$ | 300–600 °C; water/glycerol ratio of 3–9; 35,000–70,000 mL·g$^{-1}$·h$^{-1}$ | >99 | 75 | [58] |

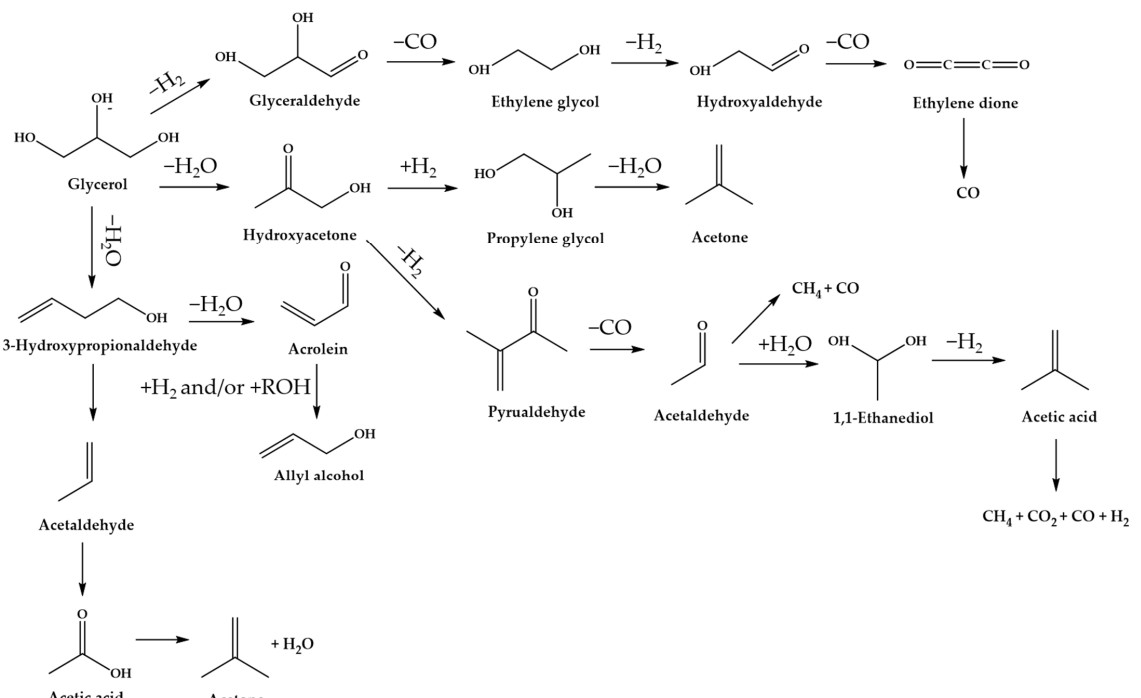

**Figure 2.** The reaction mechanism for steam reforming of glycerol [34].

Steam reforming is regarded as the most promising approach to produce $H_2$ from glycerol, and the reactions involved are shown below:

$$\text{Overall reaction: } C_3H_8O_3 + 3H_2O \rightarrow 7H_2 + 3CO_2 \tag{1}$$

$$\text{+Glycerol decomposition: } C_3H_8O_3 \rightarrow 3CO + 4H_2 \tag{2}$$

$$\text{Water-gas shift reaction: } CO + H_2O \leftrightarrow H_2 + CO_2 \tag{3}$$

Theoretically, one mole of glycerol can lead to the formation of seven moles of $H_2$; whereas the occurrence of side reactions such as methanation of CO (Equation (4)) and $CO_2$ (Equation (5)) demonstrates a negative impact on $H_2$ yield.

$$CO + 3H_2 \rightleftharpoons CH_4 + H_2O \tag{4}$$

$$CO_2 + 4H_2 \rightleftharpoons CH_4 + 2H_2O \tag{5}$$

In addition to the methanation of CO and $CO_2$, a series of other side reactions might also occur, such as dry reforming of $CH_4$ (Equation (6)), steam reforming of $CH_4$ (Equation (7)), hydrogenolysis of glycerol (Equation (8)), the Boudouard reaction (Equation (9)), methane cracking (Equation (10)), and reduction of CO (Equation (11)) and $CO_2$ (Equation (12)).

$$CH_4 + CO_2 \rightleftharpoons 2CO + 2H_2 \tag{6}$$

$$CH_4 + H_2O \rightleftharpoons CO + 3H_2 \tag{7}$$

$$C_3H_8O_3 + 2H_2 \rightleftharpoons 2CH_4 + CO + 2H_2O \tag{8}$$

$$2CO \rightleftharpoons CO_2 + C \tag{9}$$

$$CH_4 \rightarrow 2H_2 + C \tag{10}$$

$$CO + H_2 \rightarrow H_2O + C \tag{11}$$

$$CO_2 + H_2 \rightarrow 2H_2O + C \tag{12}$$

### 2.1. Transition Metals

To improve $H_2$ yield, a range of catalysts have been applied in the steam reforming of glycerol. In general, owing to their low price and wide availability, transition metals (i.e., Ni and Co)-based catalysts have been extensively investigated. On the industrial scale, Ni is most commonly used in steam reforming because of its superior catalytic performance, superior intrinsic activity, and ease of dispersal over the catalyst support. In a previous study, Charisiou et al. [59] conducted glycerol steam reforming over $Ni/Al_2O_3$, $Ni/ZrO_2$, and $Ni/SiO_2$ at 400–750 °C, and $Ni/SiO_2$ was identified as the best catalyst for $H_2$ production and demonstrated the highest level of catalytic stability. Karakoc et al. [60] also employed various Ni-based catalysts (i.e., $Ni/Al_2O_3$, $Ni/SiO_2$, and $Ni/CeO_2$) in glycerol steam reforming to produce $H_2$, and the highest $H_2$ yield of 4.82 $mol/mol_{glycerol}$ was attained at 650 °C, a Ni loading of 15 wt %, and water-to-glycerol ratio of 15. However, the deactivation of Ni-based catalysts is normally observed due to sintering and coke deposition. One solution to tackle this challenge is to integrate Ni with other transition metals such as Cu and Co. A positive synergistic effect is expected to exist between Ni and Cu or Co, which could be related to the formation of Ni-Cu or Ni-Co alloys. This formed Ni-Cu or Ni-Co alloy helps to modify Ni nanoparticles either geometrically or electronically through the formation of small ensembles of Ni sites, where the strong catalytic activity of Ni for C-C bond cleavage is maintained and coke deposition and methanation are restrained [61,62]. Additionally, the existence of Ni-Cu or Ni-Co alloys is beneficial to retarding Ni sintering when considering the relatively lower Tammann temperature of Ni (i.e., 590 °C) than the common operating temperature in steam reforming [63]. As illustrated in Figure 3, Cu is beneficial to the water–gas shift reaction, and Ni promotes

the cleavage of C–C bonds of glycerol. The hydroxyl free radicals present in the solution together with the released carbonyl radicals from the decarbonylation of glycerol can be easily absorbed by the surface of the catalyst, and thus positively proceed the water gas shift reaction [64]. Co was also utilized by Sanchez and Comelli [65] to prepare Ni-Co/$Al_2O_3$ bimetallic catalysts used in the steam reforming of glycerol to enhance $H_2$ production. In addition to promoting $H_2$ formation, Co addition is able to minimize coke formation since it offers high oxygen affinity and, hence, facilitates the sorption of oxygen species in Ni-Co. Another solution for ensuring low coke deposition and promoting $H_2$ production is to modify Ni-based catalysts with metal oxide promoters with redox and basic properties such as MgO and $La_2O_3$ [62]. Sánchez et al. [66] studied the catalytic performance of Ni supported on La-modified $Al_2O_3$ in $H_2$ production via glycerol steam reforming, and compared the performance with Ni/$Al_2O_3$ and $La_2O_3$/$Al_2O_3$. They observed that the catalyst support modified by La provided higher surface area and lower carbon deposition, thereby ensuring higher stability of Ni/$La_2O_3$-$Al_2O_3$ in the reaction. Charisiou et al. [67] modified $Al_2O_3$ supported by CaO-MgO, and a highly selective and stable catalyst, i.e., Ni/CaO-MgO-$Al_2O_3$, was synthesized; this newly developed catalyst exhibited smaller Ni species crystalline size, higher basicity, and increased surface amount of $Ni^0$ phase, which, in turn, promoted the water gas shift reaction to form $H_2$ and $CO_2$ and retarded CO production. In addition, the use of CaO-MgO as a modifier was not only advantageous for minimizing the carbon deposition on the surface of the catalyst but also altering the structure of the carbon to become less graphitic and more defective.

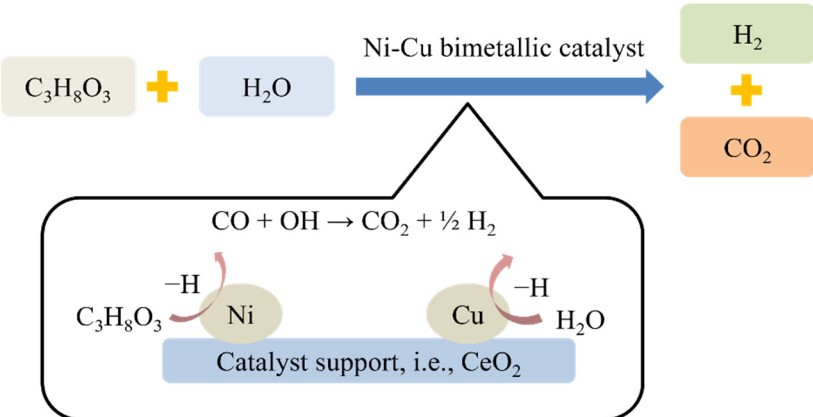

**Figure 3.** The underlying mechanism of Ni-Cu/$CeO_2$ in the steam reforming of glycerol [64].

Co, as another transition metal, has also been broadly used in the steam reforming of glycerol for $H_2$ production. Similar to Ni-based catalysts, Co-based catalysts also suffer from metal sintering, coke deposition, and catalyst deactivation. Dobosz et al. [68] applied Co/$Ca_{10}(PO_4)_6(OH)_2$ and Co-Ce/$Ca_{10}(PO_4)_6(OH)_2$ in glycerol steam reforming to produce $H_2$, and the incorporation of $CeO_2$ effectively prevented Co sintering, thus leading to a relatively higher catalyst stability and $H_2$ selectivity than Co/$Ca_{10}(PO_4)_6(OH)_2$. Adding Cu to the Co-based catalyst to suppress carbon deposition was investigated by Moogi et al. [40]. Based on the results from $H_2$-TPR analysis, the recyclability of the Co-based catalyst was enhanced in the presence of Cu, and a shift in the reduction profile toward a lower temperature was achieved. As a comparison, 5 wt % Cu-20 wt % Co/MgO led to complete glycerol conversion and the highest $H_2$ yield of 74.6%, which could be due to the smaller particle size and higher surface area and metal dispersion. They found that an acceptable catalytic activity of Cu modified catalyst was attained up to 30 h of reaction by limiting carbon formation. Menezes et al. [69] synthesized Co catalysts by the wet impregnation preparation method using three different catalyst supports: $Al_2O_3$, $Nb_2O_5$, and $Al_2O_3$-$Nb_2O_5$. Their catalytic performance in the glycerol steam reforming at 500 °C for 30 h, 20 vol % of glycerol loading, and GHSV of 200,000 $h^{-1}$ was assessed. As expected, Co/$Al_2O_3$-$Nb_2O_5$ was identified as the best catalyst with respect to the highest glycerol

conversion of 90% and $H_2$ yield of 65% at 8 h of reaction; however, coke formed in all tested catalysts after 24 h of reaction. In addition to $Nb_2O_5$, the effect of the incorporation of MgO with the Co-based catalysts on the catalytic acidity, reducibility, and cobalt dispersion was also evaluated by Menezes et al. [43]. Despite the use of $Co/MgO\text{-}Al_2O_3$ resulting in a higher glycerol conversion and $H_2$ yield, the nature of coke formed during the reaction was altered toward a filamentous rather than an amorphous structure.

### 2.2. Noble Metals

Compared to transition-metal-based catalysts, the catalysts based on noble metals (e.g., Rh, Ru, Pt, and Pd) are more stable and active in the steam reforming of glycerol for producing $H_2$. Together with Ni, Ru and Pt are regarded as the most promising metals in the steam reforming of glycerol for enhancing $H_2$ production. Until now, the catalytic activity, stability, and reducibility of Pt-based catalysts in glycerol steam reforming, such as Pt-$Ni/MgAl_2O_4$ [47], $Pt/CeO_2\text{-}Al_2O_3$ [49], $Pt/SiO_2\text{-}C$ [50], Pt-Sn/C [52], and Pt-Mn/AC [70] have been substantially explored owing to their excellent selectivity toward C-C bond cleavage. The superior catalysts utilized in the glycerol steam reforming should meet certain criteria including (i) an appropriate interaction between metal and support to ensure excellent stability and reproducibility during the reaction; (ii) high metal dispersion; and (iii) resistance to carbon deposition and sintering. Therefore, the selection of a proper support that can enhance the dispersion of active metal particles and the interaction with metal plays an important role in determining catalytic performance [32]. Recently, Buffoni et al. [50] examined the effects of catalyst support (i.e., C, $SiO_2$, and $SiO_2\text{-}C$) on the steam reforming of glycerol over Pt-based catalysts at 450 °C with respect to catalytic activity and stability. Unlike conventional oxides supports such as $SiO_2$ and $Al_2O_3$ that offer good metal–support interaction, the adoption of activated carbon as the catalyst support exhibited a high surface area, surface-enriched functional groups, and ease in metal recovery. The results suggested that $Pt/SiO_2\text{-}C$ and Pt/C led to a higher glycerol conversion of 83% and 85%, respectively, and to a $H_2$ selectivity of 51% and 52%, respectively, than those obtained using $Pt/SiO_2$ (glycerol conversion of 64% and $H_2$ selectivity of 38.8%), due to the lowest metallic dispersion observed in the presence of $Pt/SiO_2$. In terms of catalytic stability, $Pt/SiO_2\text{-}C$ was identified as the most stable catalyst over 66 h on stream, during which only 10% of its original catalytic activity was lost after the reaction, which is attributed to the better interaction between Pt and $SiO_2\text{-}C$, which thus ensures its high resistance to metal sintering. In addition to the ability to avoid sintering, the use of $SiO_2\text{-}C$ as the support was capable of deterring coke formation induced by dehydration and condensation due to the lack of strong acid sites on the surface. In another study, Manfro et al. [71] prepared Ni catalysts supported on $Al_2O_3$, $CeO_2$, and $ZrO_2$, which were employed in $H_2$ production from glycerol by steam reforming. The resulting $H_2$ selectivity in decreasing order was as follows: $ZrO_2 > Al_2O_3 \approx CeO_2$. In addition to the use of various supports, adding promoters can further improve the performance of Pt-based catalysts in glycerol steam reforming. Pastor-Pérez and Sepúlveda-Escribano [52] used Sn as the promoter in the preparation of the Pt-based catalysts; the influence of Sn addition on the activity, $H_2$ selectivity, and stability in the glycerol steam reforming was investigated. As suggested by XPS and TPR-$H_2$ analyses, the interaction between Sn and Pt was strong and demonstrated close proximity. TEM analysis indicated that the degree of metal particle agglomerations that occurred in the bimetallic catalysts was lower than in the catalyst without adding Sn. It was also found that an increase in the Sn amount in the catalyst synthesis led to less-evident particle agglomeration. Consequently, better catalytic performance in terms of $H_2$ selectivity and stability was attained using bimetallic catalysts, i.e., Pt-Sn/C, compared with a monometallic Pt/C catalyst. The higher $H_2$ selectivity obtained using Pt-Sn/C was due to the promoted CO oxidation to form $H_2$ in the presence of Sn. Furthermore, Sn was found to deter coke deposition and inhibit sintering, thereby enhancing catalytic stability during glycerol steam reforming.

In addition to Pt-based catalysts, Ru-, Rh-, and Pd-based catalysts are exceptional catalysts for the steam reforming of glycerol for $H_2$ production owing to their excellent catalytic performance and physicochemical characteristics, and their outstanding ability to deter coke deposition [32]. Senseni et al. [51] prepared different noble-based catalysts including $Rh/MgO-Al_2O_3$, $Ru/MgO-Al_2O_3$, and $Pt/MgO-Al_2O_3$ by wet impregnation, and observed that $Rh/MgO-Al_2O_3$ demonstrated the highest glycerol conversion and $H_2$ selectivity at 300–600 °C, a water-to-glycerol ratio of nine, and a GHSV of 35,000 mL·$g^{-1}$·$h^{-1}$. Additionally, the stability assessment showed that $Rh/MgO-Al_2O_3$ was the most stable catalyst for 20 h under time-on-stream by offering strong resistance to carbon deposition. Owing to their excellent catalytic activity for disrupting C-C bonds and suppressing carbon deposition, Rh-based catalysts have been employed in the steam reforming of glycerol [72]. Charisiou et al. [48] investigated the activity of $Rh/Al_2O_3$, $Rh/CeO_2-Al_2O_3$, $Rh/MgO-Al_2O_3$, and $Rh/La_2O_3-Al_2O_3$ in glycerol steam reforming at 400–750 °C, at a water-to-glycerol molar ratio of 20, and a WHSV of 50,000 mL·$g^{-1}$·$h^{-1}$. $Rh/Al_2O_3$ demonstrated the highest selectivity toward gas production and $H_2$ yield at temperatures above 550 °C; in contrast, $Rh/MgO–Al_2O_3$ was the least selective catalyst. When analyzing the chemical composition of liquid effluents, it was found that the order to stop the formation of liquid effluents was: $Rh/Al_2O_3$ at 550 °C > $Rh/La_2O_3-Al_2O_3$ at 600 °C > $Rh/CeO_2-Al_2O_3$ at 700 °C ≈ $Rh/MgO–Al_2O_3$ at 700 °C. During 12 h time-on-stream, the carbon deposited on the spent catalyst was amorphous and thus sintering was avoided, suggesting high catalytic stability during steam reforming of glycerol.

### 2.3. New Developments

### 2.3.1. Sorption-Enhanced Steam Reforming

As illustrated in Equation (1), a large quantity of $CO_2$ is generated as the by-product of glycerol steam reforming; thus, it is preferable to remove $CO_2$ in situ and, accordingly, shift the reaction toward $H_2$ formation based on Le Châtelier's Principle [48]. Thus, in order to achieve a carbon-neutral $H_2$ production process, a solid $CO_2$ sorbent such as CaO was introduced to the steam reforming process (also called sorption-enhanced steam reforming (SESR)), and the main reaction involved is: $C_3H_8O_3$ (g) + $3H_2O$ (g) + 3CaO (s) → $3CaCO_3$ (s) + $7H_2$ (g). SESR is a simple process that often leads to a high overall efficiency as production and separation are carried out simultaneously. To ensure high performance, the development of bifunctional catalysts that integrate the catalytic activity for $H_2$ production and $CO_2$ capture is a necessity [73]. Dang et al. [74] prepared a porous $Ni-CaO-Ca_{12}Al_{14}O_{33}$ bi-functional catalyst for the SESR of glycerol, and found that the $H_2$ purity was retained above 98%, with only 30% loss in the $CO_2$ sorption after 35 cycles of SESR-decarbonation. Surprisingly, the authors reported an innovative catalyst preparation method using organic molecule-intercalated layered double hydroxide (LDH) as the precursor, and the carbon species formed in situ from citrate during the calcination in an inert condition, which served as a template and a physical dispersant to deter particle aggregation (Figure 4). In addition, several bifunctional catalysts have been synthesized and tested in the SESR of glycerol to optimize $H_2$ production and reduce $CO_2$ formation [75,76]. In addition to CaO-based sorbents for improving $H_2$ production from glycerol via SESR, a range of sorbents derived from hydrotalcite, Mg-based double salts, and alkali metal-based oxides (e.g., $Li_4SiO_4$, $Li_2ZrO_3$, and $Na_2ZrO_3$) have demonstrated positive impacts on $H_2$ production in the water gas shift reaction and steam reforming of methane [77,78]; however, so far, no study has evaluated their effectiveness in the SESR of glycerol.

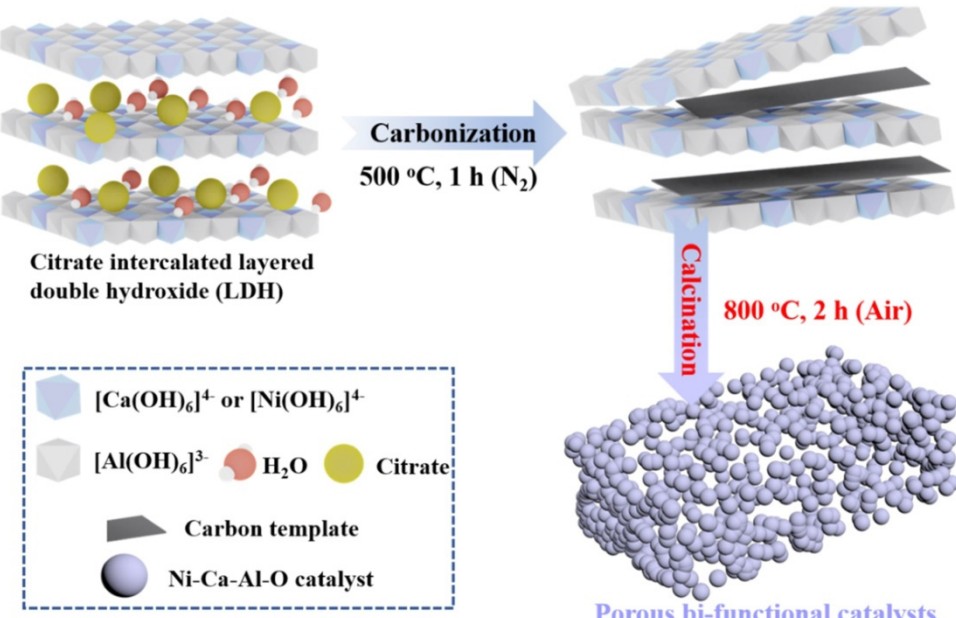

**Figure 4.** A new catalyst synthesis method for preparing a bifunctional catalyst used in the SESR reaction [74]. Copyright 2021, Elsevier.

### 2.3.2. Chemical Looping Steam Reforming

Chemical looping steam reforming (CLSR) has been applied to produce $H_2$ in a cyclic two-step process consisting of reduction and oxidation in the presence of a solid oxygen carrier (SOC), as depicted in Figure 5. CLSR allows the steam reforming process to be operated at a relatively lower temperature compared to conventional steam reforming by integrating an exothermic oxidation reaction with an endothermic reforming reaction [79]. Several studies have performed CLSR of glycerol for $H_2$ production in moving-bed re-actors [80,81] or fixed-bed reactors [82,83]. As illustrated in Figure 5, fuel is loaded into a reforming reactor, where it is oxidized by a SOC either completely to form $CO_2$ and $H_2O$ or partially to form CO and $H_2$. The glycerol conversion and product selectivity are primarily dependent on the activity and stability of the SOC, and a superior SOC should offer dual functions including (i) being readily re-oxidized by air and reduced by fuel, and (ii) providing excellent catalytic performance in steam reforming and water gas shift reactions. The most commonly used SOCs are prepared by oxygen carriers such as Fe, Mn, Co, and Cu [84] supported on porous catalyst supports such as $Al_2O_3$, $TiO_2$, $ZrO_2$, $SiO_2$, and perovskites [85].

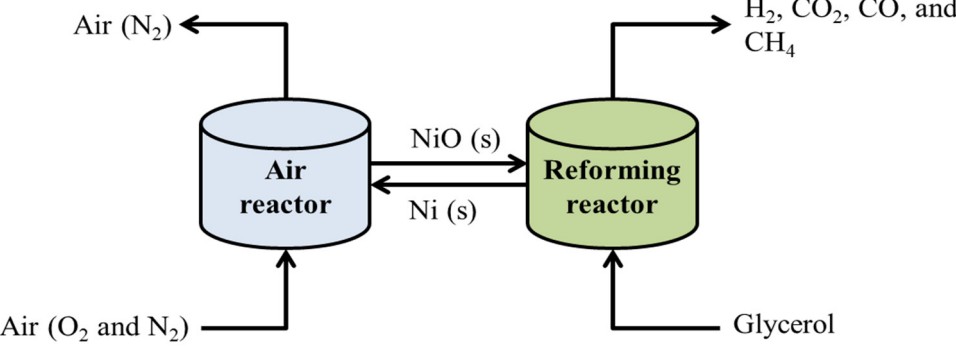

**Figure 5.** A schematic diagram for chemical looping steam reforming using NiO as the oxygen carrier [46].

### 2.3.3. Sorption-Enhanced Chemical Looping Steam Reforming

Previously, Rydén and Ramos [85] suggested combining CLSR and SESR in a one-step process (denoted as SECLR) to convert hydrocarbon to produce $H_2$ using a fluidized bed reactor and a mixture of NiO and CaO as the bed material, as shown in Figure 6. For the reforming reactor operated at a low temperature, hydrocarbon fuels are partially oxidized by the NiO and steam, as shown in Equations (13) and (14), respectively.

$$CH_4 + NiO \rightarrow CO_2 + 2H_2O + 4Ni \tag{13}$$

$$CH_4 + NiO \rightarrow CO + 2H_2 + Ni \tag{14}$$

The resulting $CO_2$ is then captured by CaO, resulting in the promotion of the water gas shift reaction, as illustrated in Equation (15). The overall reaction involved in the reforming reactor is approximately thermo-neutral.

$$CaO + CO_2 \leftrightarrow CaCO_3 \tag{15}$$

The calcination reactor is operated at intermediate temperatures and the entire process is endothermic, in which $CO_2$ is produced by $CaCO_3$ decomposition to regenerate CaO (Equation (16)).

$$CaCO_3 \leftrightarrow CaO + CO_2 \tag{16}$$

Additionally, a small flow of sweep gas consisting of $H_2O$ and $CO_2$ might be needed to enhance fluidization. In the air reactor, SOC is re-oxidized by loading air into the reactor by Equation (17), and the overall reaction involved is exothermic.

$$Ni + 1/2O_2 \rightarrow NiO \tag{17}$$

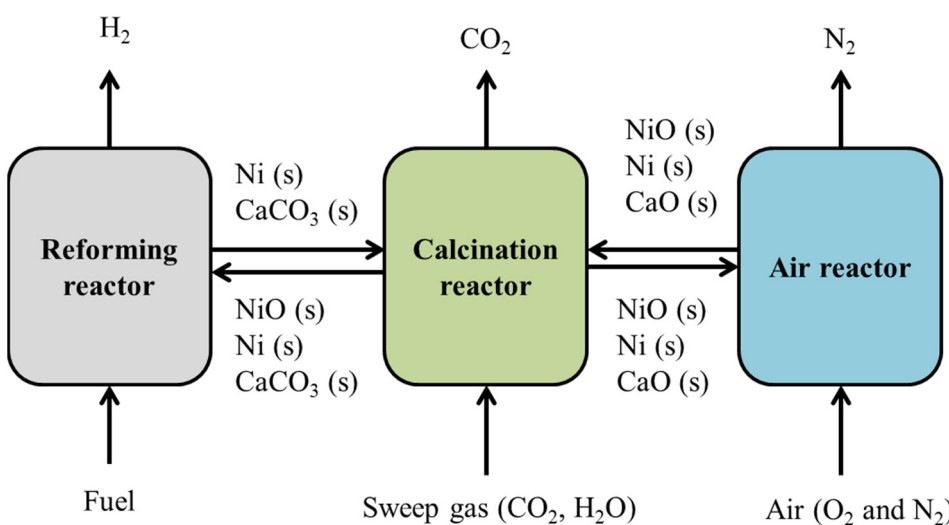

**Figure 6.** A schematic diagram for sorption-enhanced chemical looping reforming [53].

Compared to SESR, the SECLR process can be self-sufficient with heat since the $O_2$ required in the oxidation is provided by the SOC rather than steam and the following re-oxidation of SOC can produce heat. With the help of solid circulation among the reactors, the SECLR process has the potential to be operated without an external heat source for heating or cooling [79]. However, to the best of our knowledge, no study has yet performed the SECLR of glycerol to produce $H_2$, which could be an interesting direction for future research in order to develop an energy-sufficient $H_2$ production route from glycerol.

To date, a wide range of noble-metals- and transition-metals-based catalysts supported on various supports with or without a promoter have been extensively tested to produce $H_2$

from glycerol by steam reforming; however, catalyst deactivation caused by coke deposition and sintering over time is unavoidable, which consequently leads to decreases in catalytic performance and product selectivity. The detailed catalyst deactivation mechanism during the glycerol steam reforming was reviewed by Roslan et al. [23]. In particular, the cost for replacing fresh catalyst and shutdown of the industrial processes could be billions of dollars in general [23]. Despite the fact that poisoning is another cause for loss of catalytic activity, the compounds that could lead to poisoning in steam reforming are typically absent; thus, more efforts must be focused on the coke deposition and sintering of metal particles [32]. Lehnert and Claus [19] reported that the presence of NaCl in the crude glycerol led to the poisoning of metal species, thereby causing low crude glycerol conversion and fast catalyst deactivation. In addition to the new catalysts, a large amount of effort has been applied to the development of novel reactor configurations to further enhance production efficiency, including SESR, CLSR, and SECLR. For instance, SESR is capable for achieving in situ $CO_2$ removal and thus shifts the water gas shift reaction (Equation (3)) toward producing more $H_2$ gas and simultaneously limits methanation (Equation (5)) and coke formation (Equation (12)). In short, these newly developed technologies provide benefits to glycerol steam reforming by retarding the side reactions and, hence, promote $H_2$ formation [32].

Until now, although steam reforming has been the dominant conversion route to produce $H_2$, several emerging $H_2$ production technologies have been developed for glycerol such as photo-reforming and catalytic transfer hydrogenation, which still require more studies to illustrate their underlying mechanisms, to develop more efficient catalysts, and to further optimize the operating parameters for increased conversion efficiency and selectivity [23].

## 3. Utilization of Glycerol as Feedstock to Produce High-Value Chemicals

In this section, the catalytic transformation of glycerol as the feedstock to produce acrolein by dehydration, lactic acid by oxidation, and 1,3-propanediol and 1,2-propanediol via selective hydrogenolysis is discussed, with a focus on the effect of the type of active metal species and catalyst support on the product yield and selectivity, and catalyst deactivation. The reaction conditions play important roles in regulating the reaction and affecting catalytic performance. The influence of operating conditions on the glycerol conversion to acrolein, lactic acid, and propanediols was thoroughly reviewed by Belousov [86], Abdullah et al. [87].

### 3.1. Dehydration of Glycerol to Acrolein

Acrolein (also called propenal), the simplest unsaturated aldehyde, is primarily used either as a biocide in drilling water or irrigation canals to control weed and algae or as a precursor to synthesize other chemicals such as acrylic acid, glutaraldehyde, and methionine [86]. In industry, acrolein is prepared by the gas phase partial oxidation of propene over a Bi/Mo-mixed oxide catalyst. Alternatively, glycerol can be used as the feedstock to prepare acrolein via catalytic dehydration in either the gas or liquid phase. The catalysts used in the catalytic dehydration of glycerol for the production of acrolein include supported zeolites [88], heteropoly acids (HPAs) [89], mixed metal oxides [90], phosphates [91], and pyrophosphates [91]. The reaction pathways for industrial acrolein preparation and glycerol to acrolein are depicted in Figure 7. For comparison, the dehydration of glycerol in gas phase offers advantages over dehydration of glycerol in the liquid phase in terms of the ease of products' separation and a higher acrolein yield [92]. The recent studies on catalytic dehydration of glycerol for producing acrolein are summarized in Table 4.

**Figure 7.** Two reaction pathways for acrolein preparation.

**Table 4.** A summary of recent studies on catalytic dehydration of glycerol to produce acrolein.

| Catalyst | Temp (°C) | Glycerol Conversion (%) | Selectivity (%) | Reference |
|---|---|---|---|---|
| MoP | 240 | 41–50 | 87–88 | [9] |
| $H_4PMo_{11}VO_{40}$/MCM-41 | 225 | 100 | 41–68 | [93] |
| $H_3PW_{12}O_{40}$/MSU-$_x$ | 300 | 94–100 | 50–70 | [94] |
| $WO_x$/AlP; $WO_x$/ZrP; $WO_x$/TiP | 300–340 | 81–100 | 51–80 | [95] |
| $WO_3$/$ZrO_2$@SiC | 210–290 | 45–100 | 29–71 | [96] |
| HY | 250–325 | 49–68 | 38–74 | [97] |
| MOF-808 | 170 | 100 | 90 | [98] |
| SAPO-34 | 285–375 | 38–56 | 49–74 | [99] |
| HZSM-5 with modified channel lengths in the b axis | 320 | 85–99 | 80–88 | [100] |
| PW/$\gamma$-$Al_2O_3$; PMo/$\gamma$-$Al_2O_3$; SiMo/$\gamma$-$Al_2O_3$ | 280–350 | 33–94 | 11–46 | [101] |
| AlP; FeP; NiP | 280 | 89–98 | 64–82 | [102] |
| STA/$SiO_2$; HY; $SO_4$/$TiO_2$; $ZnCl_2$/$SiO_2$ | 210 | 33–94 | 76–90 | [103] |
| HZSM-5; meso-HZSM-5; CuHPO$_4$/meso-HZSM-5; $Mo_{1/3}HPO_4$/meso-HZSM-5; ZnHPO4/meso-HZSM-5; NiHPO$_4$/meso-HZSM-5; MnHPO$_4$/meso-HZSM-5 | 300 | / | Yield: 53–85 mol % | [104] |
| $H_3PW_{12}O_{40}$; Y-ASA; $H_3$PW/Y-ASA; $Ni_{0.5}H_2$PW/Y-ASA; $Ni_{1.0}$HPW/Y-ASA; $Ni_{1.5}$PW/Y-ASA | 320 | 33–82 | 48–75 | [105] |
| $Fe_{0.6}$-MFI-45-HS; $Fe_{0.6}$-MFI-60-PS; $Fe0._6$-MFI-60-IE; $Fe_{0.6}$-MFI-60-Imp; Nc-$Fe_{0.6}$-MFI-45-PS | 320 | 94–99 | 71–96 | [106] |
| Nanosheet MFI | 320 | <80–100 | 82–87 | [107] |

Typically, the reactions of solid acid catalysts involved in the dehydration of glycerol include (i) the formation of acetol on the Lewis acid sites, and (ii) the formation of acrolein on Brønsted acid sites (Figure 8) [9]. Figure 3 shows that glycerol dehydration over a solid acid catalyst often results in the formation of 3-hydroxypropanal, which is accompanied by acetol formation as the by-product. The obtained 3-hydroxypropanal further proceeds with the dehydration reaction to form acrolein. Particularly, as suggested by Chai et al. [108], Brønsted acid sites are favorable for producing acrolein, and the formation of acetol is promoted by Lewis acid sites. In the presence of strong Brønsted acid sites, coke deposition

might occur on the catalyst surface, thus leading to catalyst deactivation [109]. To reuse the spent catalyst, catalyst regeneration by burning off the coke can be carried out either continuously in situ or periodically ex situ [110]. Thus, the amount and strength of the acidic sites are essential to the catalytic dehydration of glycerol to produce acrolein with respect to unwanted side reactions and coke formation. For example, Viswanadham et al. [96] applied Keggin-type vanadium-containing phosphomolybdic acid ($H_4PMo_{11}VO_{40}$) supported on a mesoporous molecular sieve (i.e., MCM-41) in glycerol dehydration, and the results showed that acrolein selectivity was proportional to the loading of $H_4PMo_{11}VO_{40}$ up to 40 wt % and then decreased with further increases in $H_4PMo_{11}VO_{40}$ loading. Based on the results from temperature-programmed desorption of ammonia (TPD-$NH_3$), it was found that the concentration of $H_4PMo_{11}VO_{40}$ is related to the acidity of the catalyst. Similar results were reported by Ginjupalli et al. [95], where the influence of the acidic strength of a range of tungsten oxide supported on metal phosphate catalysts on the catalytic performance, reaction variables, and reactant functionalities during the gas phase glycerol dehydration for acrolein production was evaluated. In addition to the type, amount, and strength of the acidic sites, the pore size of the catalyst is another important operational variable affecting the catalytic dehydration of glycerol. The porosity and distance of internal channels of the catalyst play a significant role in the diffusion and the adsorption–desorption of the molecules, thereby affecting coke formation on the catalyst surface. Ali et al. [100] evaluated the influence of *b*-axis channel length (i.e., 60–250 nm) of HZSM-5 zeolites on the acrolein selectivity and coke formation obtained from glycerol dehydration, and the results showed that the HZSM-5 catalyst with a *b*-axis channel length of 60 nm led to the highest glycerol conversion (100%) and acrolein selectivity (88%). The shortest channel length could lead to a high availability of active sites and improved diffusion, which, in turn, drastically limit coke formation. In terms of the effect of pore size, Zhang et al. [111] observed that hierarchical-structured zeolites with diverse meso-porosity demonstrated better stability and acrolein selectivity compared with conventional zeolites with sole microporous structure. Additionally, the presence of mesopores in the zeolites is helpful for making the catalyst more tolerant to coke formation, particularly in the case of an open and interconnected mesopore architecture. As suggested by the Kelvin equation (Equation (4)), catalyst deactivation might be caused by pore condensation, and a greater degree of pore condensation can be observed in the smaller mesoporous structures [112].

**Figure 8.** The reaction pathways for glycerol dehydration over a solid acid catalyst [63].

$$RTln\left(\frac{P_c}{P_0}\right) = \frac{-2\sigma_{VL}cos\theta}{(n_L - n_V)H} \tag{18}$$

where $P_c$ and $P_0$ are the pressure at which pore condensation occurs and the saturated vapor pressure, respectively; $T$ and $R$ represent the absolute temperature and gas constant, respectively; $\sigma_{VL}$ is the vapor–liquid surface tension; $\theta$ represents the contact angle; $n_L$ and $n_V$ are the molar density of the bulk liquid and vapor phase, respectively; and $H$ is the pore width.

Undeniably, coke formation and the associated catalyst deactivation are the major challenges faced by the catalytic dehydration of glycerol for acrolein production. The possible reaction pathways for coke formation during glycerol conversion to acrolein are shown in Figure 9. In addition to the modification of the pore size of the catalyst, as earlier discussed in this section, doping the catalyst with noble metals (e.g., Ru, Pt, or Pd) is another solution [113]. Doping Ru, Pd, or Pt into the catalyst together with $H_2$ addition are effective in preventing coke formation through the hydrogenation of the coke precursors, consequently extending the catalyst's lifetime [113]. Trakarnpruk [114] reported that Pt doping in $H_3PW_{12}O_{40}$/Zr-MCM-$_{41}$ catalyst was capable of suppressing coke formation and dramatically enhancing catalyst stability. In another study, Ma et al. [115] synthesized and tested $H_3PW_{12}O_{40}$/MCM-$_{41}$, $H_3PW_{12}O_{40}$/Zr-MCM-$_{41}$, and Pd-$H_3PW_{12}O_{40}$/Zr-MCM-$_{41}$ in the catalytic dehydration of glycerol for acrolein production, and the results indicated that Pd doping did not alter the mesoporous structure of the catalyst but decreased the specific surface area, pore volume, and pore size. Although no significant change was observed in the total acidity of the catalyst, the amount of Brønsted acid sites increased with a decrease in the amount of Lewis acid sites. Additionally, the use of Pd-$H_3PW_{12}O_{40}$/Zr-MCM-$_{41}$ led to the highest glycerol conversion of 94% and acrolein selectivity of 85%, which were accompanied by higher catalyst stability over 50 h of reaction compared to $H_3PW_{12}O_{40}$/Zr-MCM-$_{41}$. The third approach to mitigate coke formation is co-feeding oxygen or air through the oxidation of coke precursors to form CO and $CO_2$. Nadji et al. [90], for example, studied the effect of $O_2$ on acrolein production from glycerol. When co-feeding $O_2$, the glycerol conversion and acrolein selectivity were 100% and 85%, respectively, during 8 h of reaction. Conversely, a significant decrease in the glycerol conversion from 99% to 55% was found after 8 h of reaction, along with a relatively lower acrolein selectivity ranging from 70% to 85%. Based on the results, the authors speculated that co-feeding $O_2$ in the glycerol dehydration is not only helpful for preventing the formation of coke but also suppressing acetol formation. Similar results were observed by Dalil et al. [116], where the dehydration of glycerol was performed over $WO_3$/$TiO_2$ in 10 mol % $O_2$/Ar at 280 °C. However, the presence of $O_2$ promotes the formation of carboxylic acids (e.g., formic acid, acetic acid, and acrylic acid), which could be attributed to the oxidation of aldehydes [5]. Recently, Xie et al. [99] used microwave heating in the catalytic glycerol dehydration to prevent coke formation. Their newly designed microwave system consists of feedstock storage, peristaltic pump, preheater, quartz reactor, catalyst bed, microwave oven, infrared irradiation thermometer, temperature controller, liquid product storage, and water seal. The authors reported that the glycerol conversion (83.8%) and acrolein selectivity (53.5%) obtained from electric heating were lower than those obtained from microwave heating at 250 °C (glycerol conversion: 100% and acrolein selectivity: 71.1%), which might be due to the differences in the heating mechanism between microwave heating and conventional heating. In conventional heating, heat is transferred from the reactor wall to the interior of the catalyst bed by conduction, resulting in a lower temperature at the interior of the catalyst bed. In microwave heating, the electromagnetic energy absorbed by the material is directly converted into heat at the molecular level. The major challenges for applying microwave irradiation in glycerol dehydration include safety issues, the lack of an accurate temperature measuring device, and the difficulty of selecting construction materials.

To date, many efforts have been aimed at suppressing coke formation and prolonging catalyst lifetime for glycerol dehydration to produce acrolein through (i) doping noble metals, (ii) modifying the porosity and channel length, and (iii) co-feeding $O_2$; however, the catalysts will eventually be deactivated. Thus, catalyst regeneration by burning off the coke in air or $O_2$ has been carried out through in situ regeneration [117] or periodic regeneration [118], but explosion might occur under high $O_2$ concentration (i.e., 7%) [113]. It is essential for future studies to design and develop innovative reactor configurations for the catalyst regeneration used in the catalytic dehydration of glycerol for producing acrolein.

**Figure 9.** Possible reaction pathways for coke formation during catalytic dehydration of glycerol to produce acrolein [113]. Copyright 2021, Elsevier.

### 3.2. Oxidation of Glycerol to Lactic Acid

Lactic acid is an essential ingredient in the food industry as an acidulant or inhibitor for bacterial spoilage, in the textile industry as a mordant to improve color durability, in the cosmetic industry as a moisturizer, and in the dairy industry as a pH regulator, as well as an important monomer for manufacturing biopolymer polylactic acid (PLA) [119]. Traditionally, lactic acid is synthesized through sugar fermentation using carbohydrates as the carbon substrate; this method suffers from poor productivity and expensive operating costs due to the high cost of the enzyme, complex post-treatment by purification, and low scalability. As such, recent studies have employed glycerol as the feedstock to produce lactic acid via oxidation, as summarized in Table 5. The proposed reaction mechanism for converting glycerol to lactic acid via oxidation is depicted in Figure 10. As illustrated in Figure 10, glycerol conversion to lactic acid follows these steps:

i.  Glycerol is initially dehydrogenated to glyceraldehyde in the presence of homogenous base and metal sites;
ii. Dehydration of glyceraldehyde is followed by keto-enol tautomerism to form pyrualdehyde;
iii. The formed pyrualdehyde is converted to lactic acid by an intra-molecular Cannizzaro reaction.

**Table 5.** A summary of recent studies on glycerol oxidation to produce lactic acid.

| Catalyst | Temp (°C) | Glycerol Conversion (%) | Selectivity (%) | Reference |
|---|---|---|---|---|
| Au-Pt/$Al_2O_3$ | 70–85 | 3.3–49 | 2.0–47 | [11] |
| $Co_3O_4$/$CeO_2$; $Co_3O_4$/$ZrO_2$; $Co_3O_4$/$TiO_2$ | 250 | 49–59 | 68–90 | [120] |
| Au/ZSM-11; Pt/ZSM-11; Pd/ZSM-11; Au-Pt/ZSM-11; Au-Pd/ZSM-11 | 70 | 27–66 | 27–45 | [121] |
| Pt/AC; Pd/AC | 230 | 70–86 | 99–100 | [122] |
| Au/bentonite | 90 | 82 | 92 | [123] |
| Au-Pt/$TiO_2$-P25; Au-Pt/$TiO_2$-NC; Au-Pt/$TiO_2$-A; Au-Pt/$TiO_2$-R | 110 | 3–99 | 31–84 | [124] |
| Au-Pt/$TiO_2$ | 40–120 | 24–100 | 10–72 | [125] |
| Ni-NiO$_x$/C-200; Ni/C; NiO; Ni@C; Ni-NiO$_x$@C-200 | 200 | 20–100 | / | [126] |
| Cr/ZSM-11; Cu/ZSM-11 | 60 | 24–43 | 5–68 | [127] |

**Table 5.** *Cont.*

| Catalyst | Temp (°C) | Glycerol Conversion (%) | Selectivity (%) | Reference |
|---|---|---|---|---|
| Zr-Ce/SBA-15 | 240–280 | 59–81 | / | [128] |
| Pd$_3$/HAP; Pd$_{0.75}$/HAP; Pd$_{1.5}$/HAP; HAP | 230 | 16–99 | 2–90 | [129] |

**Figure 10.** The reaction pathways for converting glycerol to lactic acid by oxidation [126]. Copyright 2021, Elsevier.

Alkali is capable of catalyzing dehydrogenation and dehydration reactions when subjected to hydrothermal conditions; thus, it has been extensively applied in lactic acid production from glycerol. For example, Zhang et al. [130] performed selective oxidation of glycerol over Pt/AC to produce lactic acid in different basic solutions including LiOH, NaOH, KOH, and Ba(OH)$_2$, and the results showed that the order for lactic acid selectivity was as follows: LiOH > NaOH > KOH > Ba(OH)$_2$. The highest selectivity of lactic acid (69.3%) was achieved at 90 °C for 6 h and a LiOH-to-glycerol molar ratio of 1.5, which achieved a glycerol conversion of 100%. Despite the use of Pt/AC demonstrating excellent catalytic stability in glycerol oxidation, it had a detrimental influence on the conversion of the intermediate toward lactic acid formation by shifting the reaction to form glyceric acid. Yang et al. [131] observed that 100% glycerol conversion and 94.6% lactic acid selectivity were achieved when conducting oxidation in NaOH solution, at 180 °C, for 8 h, and at 1.4 MPa of N$_2$. Similarly, Yin et al. [132] tested different Cu-based catalysts (i.e., Cu/hydroxyapatite, Cu/MgO, and Cu/ZrO$_2$) for glycerol conversion to lactic acid in NaOH solution. They observed that both Cu/hydroxyapatite and Cu/MgO exhibited better catalytic performance than Cu/ZrO$_2$, which was mainly due to the differences in the basicity among Cu-based catalysts. A maximum selectivity of lactic acid of 90% was obtained at 230 °C for 2 h and at a NaOH concentration of 1.1 mol/L, along with a 91% of glycerol conversion. Notably, glycerol conversion in alkaline solution is usually operated at severe conditions (i.e., temperature of 280–290 °C) as dehydrogenation is an energy-demanding process. Nevertheless, alkali-assisted C-C bond cleavage might be simulated in harsh conditions, thereby leading to the formation of a series of undesirable by-products such as acetic acid, acrylic acid, formic acid, and oxalic acid. Consequently, to limit C-C bond cleavage forming undesired by-products, dehydrogenation must be performed at moderate conditions, which can be achieved using noble-metals-based catalysts (e.g., Pt and Pd). Feng et al. [133] synthesized a Pt-based catalyst (i.e., Pt/L-Nb$_2$O$_5$) and used it in the glycerol conversion for producing lactic acid under base-free conditions. The results showed that the presence of Lewis acid sites of Pt/L-Nb$_2$O$_5$ was helpful for the transformation of pyruvic aldehyde to lactic acid. The authors also reported that the selectivity of lactic acid and glycerol conversion were 91% and 81%, respectively, when using Pt/L-Nb$_2$O$_5$ as the catalyst. Marques et al. [134] investigated glycerol conversion to lactic acid over Pd/C, and a 99% glycerol conversion and a 46% lactic acid selectivity were

obtained. In addition to Pd and Pt, the efficiency of using Au-Pt or Au-Pd alloy in lactic acid production from glycerol oxidation was also assessed because: (i) Au is an effective catalyst for alcohol oxidation by molecular oxygen in the liquid phase; (ii) Au commonly shows excellent catalytic performance and is highly resistant to catalyst deactivation. Thus, several previous studies were carried out to apply Au-Pt/C [135], Au-Pt/TiO$_2$ [136], or Au-Pt/CeO$_2$ [137] in glycerol oxidation to enhance lactic acid production. In general, the use of bimetallic catalysts demonstrated improved catalytic performance in terms of glycerol conversion and lactic acid selectivity compared with monometallic catalysts [137]. However, noble-metals-based catalysts tend to be deactivated and show poor recyclability, which might result from over-oxidation, metal leaching and sintering, and poisoning by molecular oxygen. Catalyst deactivation could be limited, to some extent, by purging N$_2$ into the catalyst support [138]. To enhance catalytic reducibility, several strategies have been developed [119]:

i. Pt and Pd supported by activated carbon showed great stability during glycerol oxidation to produce lactic acid;
ii. The development of bimetallic catalysts supported on CeO$_2$ where insignificant loss in the catalytic activity was observed upon recycling five times;
iii. Adding non-noble metal promoters.

Overall, to date, a large gap remains in the development of excellent heterogeneous catalysts for the oxidation of glycerol for producing lactic acid; thus, more work is required.

### 3.3. Selective Hydrogenolysis of Glycerol to 1,3-Propanediol

1,3-propanediol has been extensively applied in the synthesis of polymers such polyethers, polyurethanes, and polyesters; most importantly, polypropylene terephthalate (PPT) fibers can be manufactured based on 1,3-propanediol and terephthalic acid [139]. During 1,3-propanediol formation, it is essential to selectively break down the secondary C-O bond of glycerol, which still remains a big challenge because of the similarities in the activation energies among the three C-O bonds of glycerol, thus complicating the discrimination. Additionally, the accessibility of the secondary C-O bond is restricted due to steric hindrance [140]; it was reported that a range of by-products (e.g., 1-propanol and 2-propanol) can be generated in an excessive hydrogenolysis [141]. As shown in Figure 11, glycerol hydrogenolysis to produce 1,3-propanediol occurs on Brønsted acid sites at high a hydrogen pressure via Route 1. At higher temperatures, more glycerol can be converted into 3-hydroxypropionaldehyde (3-HPA) as an intermediate, while the formation of acrolein and monoalcohols can be stimulated. Thus, the operating temperature applied in the hydrogenolysis of glycerol for 1,3-propadeniol is usually below 200 °C. In addition to Brønsted acid sites, the hydrogenolysis reaction can also occur at Lewis acid sites where 1,2-propanediol formation is promoted by Route 2. Glycerol is initially dehydrated to form hydroxyacetone, which further undergoes hydrogenation into 1,2-propanediol [142].

**Figure 11.** General reaction routes involved in the hydrogenolysis of glycerol [142].

Until now, a great deal of effort has been directed to the design of an effective catalyst that can improve the selectivity toward 1,3-propanediol formation, among which Pt-W-based catalysts have recently been broadly investigated due to their suitable activity in the selective formation of 1,3-propanediol and potential industrial applications. Specifically, Pt-W-based catalysts offer dual roles including (i) active metal Pt species promoting the activation of $H_2$ to provide the hydrogen source and (ii) Brønsted acid sites forming when the hydrogen species reach $WO_x$ species through hydrogen spillover, which is responsible for activating glycerol [141,142]. Edake et al. [143] investigated the hydrogenolysis of glycerol over Pt-$WO_3$/$Al_2O_3$ at 240–300 °C and ambient pressure in a fluidized bed reactor, and the highest glycerol conversion and 1,3-propanediol selectivity of 99% and 14%, respectively, were attained at 260 °C, a $H_2$-to-glycerol ratio of 28, and a WHSV of 0.14 $h^{-1}$. $Al_2O_3$ is an efficient catalyst support that provides a higher surface area and serves as an anchor to fix glycerol on the surface, so has been regarded as one of the most effective supports used in glycerol hydrogenolysis to produce a high yield and selectivity of 1,3-propanediol, e.g., selectivity and yield of 66% and 42%, respectively, as reported by Zhu et al. [144]. In addition to $Al_2O_3$, a variety of catalyst supports have been tested such as $SiO_2$ [145], $ZrO_2$ [146], $WO_3$ [147], $AlPO_4$ [148], SBA-15 [149], and AlOOH [150]. In a study, Priya et al. [95] evaluated the influence of the catalyst support on glycerol conversion into 1,3-propanediol at 260 °C, 10 wt % of glycerol loading, and a 0.1 MPa of $H_2$ initial pressure, and the tested supports included $ZrO_2$, sulfated $ZrO_2$, $Al_2O_3$, $AlPO_4$, activated carbon, and Y-Zeolite. Among them, $AlPO_4$ was identified as the best catalyst support, resulting in a glycerol conversion of 100% and a 1,3-propanediol selectivity of 35.4%. The results suggested that the existence of weak acid sites play a positive role in the formation of 1,3-propanediol, and a high strength of weak acid sites was found in Pt/$AlPO_4$ based on the $NH_3$-TPD analysis, ensuring the highest catalytic performance during the hydrogenolysis of glycerol for 1,3-propanediol production. In another study, Zhu et al. [151] modified $TiO_2$-supported Pt-$WO_x$ catalyst by sulfate doping and then employed it in 1,3-propanediol production via hydrogenolysis since sulfate doping was

reported to be helpful for increasing the surface area of the catalyst and improving metal dispersion [152]. The results showed that sulfate-doped Pt-WO$_x$/S-TiO$_2$ achieved 100% glycerol conversion and 36% 1,3-propanediol selectivity at 120 °C and 4 MPa. Conversely hand, nonsulfate-doped Pt-WO$_x$/TiO$_2$ led to a significantly lower glycerol conversion of 57% but a higher selectivity of 66% toward 1,3-propanediol formation. This could be due to the over-strong hydrogenolysis activity of sulfate-doped catalyst. Subsequently, the addition of sulfate-doped Pt-WO$_x$/S-TiO$_2$ was found to achieve a glycerol conversion of 75% and a 1,3-propanediol selectivity of 47% when lowering the severity of the reaction, i.e., temperature of 100 °C, implying the beneficial impacts of sulfate on hydrogenolysis. This high hydrogenolysis efficiency offered by sulfate doping is related to the better dispersion of Pt and WO$_x$ species and higher concentrations of Brønsted acid sites. Additionally, a better catalytic stability was detected when using the sulfate-doped Pt-WO$_x$/S-TiO$_2$ as the catalyst, i.e., the glycerol conversion reduced from 100% to 83% and 82% at the third and fourth run, respectively, resulting from stronger antileaching in the presence of sulfate doping. Nevertheless, several challenges remain [141]:

i.   A deep understanding of the synergistic influence between Pt species and WO$_x$ species is needed;
ii.  Characterization tools to elucidate the correlation between glycerol activation and the surface and structure properties of catalyst are lacing;
iii. More effective catalysts must be developed for the selective hydrogenolysis of glycerol by different modification and catalyst preparation methods.

Apart from Pt-W-based catalysts, Ir-Re-based catalysts, as other widely investigated catalysts, have also been investigated to enhance 1,3-propanediol selectivity and yield from the hydrogenolysis of 1,3-propanediol [153–157]. Chanklang et al. [61], for instance, explored the effectiveness of using Ir-ReO$_x$/H-ZSM-5 in glycerol hydrogenolysis to produce 1,3-propanediol at 180–240 °C and 2–8 MPa of H$_2$ for 1–8 h, and the incorporation of Re led to better Ir dispersion. They also found that a maximum yield of 1,3-propanediol of 2.8% with a glycerol conversion of 14.9% at 220 °C and 4 MPa of H$_2$ for 8 h. Liu et al. [154] synthesized an Ir-ReOx catalyst supported on SiO$_2$, and the results showed that the highest 1,3-propanediol yield and selectivity were 32% and 47%, respectively, at 120 °C and 8 MPa of H$_2$ for 24 h, which was accompanied by a glycerol conversion of 69%. In addition to H-ZSM-5 and SiO$_2$, KIT-6, as another ordered mesoporous silica with a cubic arrangement of interconnected pores, was also investigated in the preparation of an Ir-ReO$_x$-based catalyst to enhance catalytic performance in glycerol hydrogenolysis for 1,3-propanediol production [157]. Re is a rare earth element, which significantly hinders its large-scale application. The most recent investigations into the hydrogenolysis of glycerol to produce 1,3-propanediol in the presence of either Pt-W-based catalysts or Ir-Re-based catalysts are summarized in Table 6.

**Table 6.** A summary of recent studies on glycerol hydrogenolysis to 1,3-propanediol.

| Catalyst | Temp (°C) | Glycerol Conversion (%) | Selectivity (%) | References |
|---|---|---|---|---|
| Pt-WO$_x$/0.5MCF; Pt-WO$_x$/1.0MCF; Pt-WO$_x$/1.5MCF; Pt-WO$_x$/2MCF; Pt-WO$_x$/2.5MCF | 150 | 37–100 | 61–66 | [17] |
| Pt-WO$_x$/Al$_2$O$_3$ (rod-like); Pt-WO$_x$/Al$_2$O$_3$ (flake-like); Pt-WO$_x$/Al$_2$O$_3$ (spindle-like) | 160 | 19–80 | 46–50 | [142] |

**Table 6.** *Cont.*

| Catalyst | Temp (°C) | Glycerol Conversion (%) | Selectivity (%) | References |
|---|---|---|---|---|
| Ir-ReOx/H-ZSM-5; Ir-ReO$_x$/TiO$_2$; Ir-ReO$_x$/SiO$_2$ | 200 | 2–6 | 13–34 | [153] |
| Pt/Al$_2$O$_3$; Pt-WO$_x$/Al$_2$O$_3$ | 140 | 23–36 | / | [158] |
| Pt/SiO$_2$; Pt-WO$_x$/SiO$_2$ | 180 | 16–64 | 48–57 | [159] |
| Pt-WO$_x$/Al$_2$O$_3$ | 180 | 58 | 40 | [160] |
| Pt-WO$_x$/SiO$_2$-Al$_2$O$_3$; SiO$_2$-Al$_2$O$_3$; Pt-WO$_x$/SiO$_2$ | 210 | 1–64 | 0–27 | [161] |
| Pt-WO$_x$/ZrO$_2$; Pt-WO$_x$/TiO$_2$; Pt-WO$_x$/ZrO$_2$-TiO$_2$ | 140 | 26–74 | 32–40 | [162] |
| Pt-WO$_x$/SAPO-34; SAPO-34; Pt/SAPO-34 | 210 | 0–48 | 6–19 | [163] |

*3.4. Selective Hydrogenolysis of Glycerol to 1,2-Propanediol*

Currently, the transformation of glycerol into 1,2-propanediol by selective hydrogenolysis has received much attention due to the ability of 1,2-propanediol as an industrial monomer to produce polyester resins, detergents, antifreeze agents, additives in paint, food, etc. Industrially, 1,2-propanediol is synthesized through the hydration of propylene [164]. When using glycerol as the feed, selective hydrogenolysis of glycerol for producing 1,2-propanediol has been extensively carried out over noble metals (e.g., Pd, Pt, and Ru) because of their capability to active hydrogen molecules. Oberhauser et al. [165] found that Pt nanoparticles supported on carbonaceous catalyst supports can catalyze glycerol hydrogenolysis at 160 and 180 °C. The used carbonaceous supports were Katjen Black EC-600JD, Vulcan XC-72, and fewer-layer graphene. Among them, Katjen Black EC-600JD exhibited the highest surface area and contributed to retarding the agglomeration of metal nanoparticles, along with showing the highest selectivity for 1,2-propanediol of 70%, achieved at 160 °C. Silveira et al. [166] prepared a Ru catalyst supported on sugarcane-straw-derived active carbon and commercial activated carbon, which was tested by conducting glycerol hydrogenolysis at 200 °C, 5 MPa, 6 h, and 10 vol % of glycerol loading. The results showed that the formation of 1,2-propanediol was more favorable than 1,3-propanediol formation under the investigated conditions, which might be due to the dominance of Lewis sites. To further enhance the performance of Ru-based catalysts in glycerol hydrogenolysis, Sherbi et al. [16] added another metal, Cu, into the preparation of Ru-based catalyst supported on carbon nanotubes (CNTs), and this developed catalyst reached 93.4% 1,2-propanediol selectivity at 200 °C, 5 MPa of H$_2$, 20 h, and 20 wt % of glycerol loading, along with glycerol conversion of 18%. Similar results were reported by Wu et al. [167], where Ru-Cu/CNT catalysts were applied in glycerol hydrogenolysis to enhance 1,2-propanediol production, and the results showed that the bimetallic catalysts showed a higher selectivity toward 1,2-propanediol formation than single-metal catalysts due to the hydrogen spillover effect resulting from the presence of highly dispersed tiny Ru clusters on the surface of Cu particles. The main benefits of using Ru-based catalysts for glycerol hydrogenolysis were reviewed [168], as indicated below:

i. The combination of active metal particles and acid support could lead to the formation of 1,2-propanediol as the main product throughout glycerol conversion, especially under mild conditions (i.e., temperature below 180 °C);

ii. The occurrence of over-hydrogenolysis in the presence of Ru-based catalysts can be identified when conducting the reaction under severe conditions (i.e., temperature above 240 °C), which causes the product formation to shift from 1,2-propanediol to 1-propanol and propane.

Pd, as another noble metal, has also been used to synthesize catalysts in the 1,2-propanediol production from glycerol. Mauriello et al. [169] studied the hydrogenolysis of glycerol at 180 °C and 5 bar without adding hydrogen gas; instead, an external hydrogen source, i.e., 2-propanol, was used as the reaction medium, and the investigated catalysts included Pd/Co, Pd/CoO, Pd/Co$_3$O$_4$, Pd/Fe, Pd/Fe$_2$O$_3$, Pd/Fe$_2$O$_3$, and Pd/SiO$_2$. The interaction between Pd and other metals (i.e., Co and Fe) can modify the electronic properties of Pd and results in the formation of bimetallic Pd-Co or Pd-Fe sites, thereby promoting catalytic performance in the hydrogenolysis of glycerol, creating a shift toward 1,2-propanediol production. In addition to noble metals, transition metals such as Ni [170], Cu [171], and Co [172] have been broadly explored in the selective hydrogenolysis of glycerol to generate 1,2-propanediol due to their low price and high activity and selectivity [173]. Among them, Cu-based catalysts are the most commonly utilized catalysts in glycerol hydrogenolysis, which is primarily related to their strong ability to cleave C-O bonds [164]. To date, Cu-based catalysts, either monometallic or bimetallic, have been designed: Cu-Zn/Al$_2$O$_3$ by Mishra et al. [171], Cu/SiO$_2$ by Shan et al. [174], Cu-Ni/Y zeolite by de Andrade et al. [175], Cu/ZnO by Wang et al. [176], Cu/metal oxides (i.e., Al$_2$O$_3$, SiO$_2$, ZnO, and MgO) by Zhou et al. [177], Cu/AlOOH by Wu et al. [178], Cu-Ni/SiO$_2$ by Lee et al. [179], Cu-Ru/TiO$_2$ by Salazar et al. [180], Cu-Ni/Al$_2$O$_3$ by Poddar et al. [181], and Co/dolomite by Azri et al. [182]. The underlying mechanism involved in Cu-catalyzed glycerol hydrogenolysis was reviewed by Montassier et al. [183]. Initially, glycerol is dehydrogenated on Cu to produce glyceraldehyde, and then the formation of 1,2-propanediol is achieved through a nucleophilic reaction of water or absorbed OH species, dihydroxylation, and hydrogenation of aldehyde (2-hydroxy acrolein) [164]. The relevant studies on the selective hydrogenolysis of glycerol for 1,2-propanediol production over either noble-metals-based catalysts or transition-metals-based catalysts are summarized in Table 7. Several studies have been performed to compare the catalytic performance amongst various noble-metals-based and transition-metals-based catalysts in 1,2-propanediol production by glycerol hydrogenolysis. For example, Kang et al. [184] prepared and characterized various bimetallic catalysts including Pt-Cu/SiO$_2$, Pd-Cu/SiO$_2$, Ag-Cu/SiO$_2$, and Ni-Cu/SiO$_2$ using a series of analytical techniques, and found that Pt-Cu/SiO$_2$ demonstrated the largest metal particles dispersion and smallest particle size. In the following hydrogenolysis investigation at 200 °C, 4 MPa of H$_2$, and 12 h, the authors reported that using Pt-Cu/SiO$_2$ led to an almost 100% glycerol conversion and the highest selectivity to 1,2-propanediol of 96% of the considered bimetallic catalysts (i.e., Pd-Cu/SiO$_2$, Ag-Cu/SiO$_2$, and Ni-Cu/SiO$_2$) and monometallic catalysts (i.e., Cu/SiO$_2$ and Pt/SiO$_2$). This was accompanied by a relatively high catalytic stability offered by Pt-Cu/SiO$_2$, which could be due to the decreased metal agglomeration tendency in the presence of Pt. Von Held Soares et al. [185] conducted glycerol hydrogenolysis over Pt/Fe$_3$O$_4$, Pd/Fe$_3$O$_4$, and Ni/Fe$_3$O$_4$, and the order of catalytic activity was as follows: Pt > Pd > Ni.

Overall, the transformation of glycerol to value-added C$_3$ chemicals (i.e., acrolein, lactic acid, 1,3-propanediol, and 1,2-propanediol) over heterogeneous catalysts is an essential component of achieving the sustainable and economic production of biodiesel. To ensure a high yield of the target products, proper catalyst design including the selection of active metal species, support material, catalyst preparation method, and the type and strength of acid sites) and reaction conditions must be carefully determined.

**Table 7.** A summary of recent studies on glycerol hydrogenolysis to produce 1,2-propanediol.

| Catalyst | Temp (°C) | Glycerol Conversion (%) | Selectivity (%) | Reference |
|---|---|---|---|---|
| Pt/Al$_2$O$_3$; In/Al$_2$O$_3$; Pt-In/A Al$_2$O$_3$; Pt/SiO$_2$; In/SiO$_2$; Pt-In/SiO$_2$ | 240 | 1–39 | 16–49 | [115] |
| Cu/MgO; Cu-Ru/MgO | 220 | 9–48 | 7–17 | [116] |
| Cu/Dol; Ni/Dol; Co/Dol; Fe/Dol; Zn/Dol; Dolomite | 200 | 9–79 | 0–79 | [170] |
| Cu/Ga$_{2.3}$-HT | 300 | 21–95 | 0–97 | [173] |
| Ni/NaY-zeolite; Cu/NaY-zeolite; Ni-Cu/NaY-zeolite | 260 | 71–96 | 13–44 | [175] |
| Ni-Cu/Al$_2$O$_3$; Ni/SiO$_2$; Ni/WO$_3$; Ni/B-Al$_2$O$_3$ | 200 | 1–67 | 0–90 | [181] |
| Ru-Cu/m-ZrO$_2$; Ru-Cu/CaO-ZrO$_2$; Ru-Cu/SO$_4$-ZrO$_2$; Ru-Cu/WO$_3$-ZrO$_2$ | 180–200 | 1–30 | 51–87 | [186] |
| Ru/K-OMS-2; Cu/K-OMS-2; Ni/K-OMS-2 | 180–220 | 32–100 | 69–91 | [187] |

## 4. General Catalyst Preparation Strategies and Associated Characterization Methods

In addition to the type of active metal and catalyst support, catalyst preparation is another important factor affecting catalytic performance and product yield and selectivity in the catalytic transformation of glycerol to fuels and chemicals [188]. The most used catalyst preparation methods in catalytic glycerol valorization include: impregnation [35], hydrothermal [160], precipitation [189], sol-gel [190], and wet incipient [191] methods. Among them, the methods commonly used for catalyst preparation are impregnation and precipitation. Impregnation is mainly dependent on the interaction between the surface of the support and the species in the prepared solution, which can be further divided into wet impregnation and dry impregnation according to the volume of impregnation solution introduced to the pores of the support material. In comparison, the main benefit offered by dry impregnation is that the amount of added components to the catalyst is easier to control than wet impregnation; however, the catalyst synthesized by dry impregnation might not be as uniform as that prepared by wet impregnation [192]. Precipitation is the most widely applied catalyst preparation method because of its low cost and simplicity. For example, several industrially used catalysts are prepared by precipitation or co-precipitation such as SiO$_2$-Al$_2$O$_3$ used in fluid catalytic cracking (FCC), Fe$_2$O$_3$ applied in the Fisher Tropsch process, and Cu-ZnO/Al$_2$O$_3$ employed in methanol synthesis [193]. To the best of our knowledge, no study has yet compared different catalysts preparation methods in terms of their activity, stability, and reducibility during glycerol transformation, which could be a future research direction. After catalyst synthesis, a series of analytical techniques is needed to determine the characteristics of the prepared catalysts, and the most widely applied characterization methods are X-ray fluorescence spectrometry, N$_2$ adsorption-desorption analysis, X-ray diffractometry (XRD), transmission electron microscopy (TEM), and temperature-programmed desorption of ammonia (NH$_3$-TPD) analysis [36,194].

## 5. Future Perspectives and Conclusions

Glycerol, as a by-product generated in an immense amount from the transesterification used to produce biodiesel, must be used in an economical and environmentally friendly

manner. To achieve this goal, various valorization technological routes have been developed including: (i) steam reforming of glycerol to produce $H_2$ and syngas; (ii) dehydration of glycerol to produce acrolein; (iii) oxidation of glycerol to produce lactic acid; and (iv) selective hydrogenolysis of glycerol to produce either 1,3-propanediol or 1,2-propanediol. The research advances and main challenges of each of the above technological routes are summarized as follows:

i.   For the steam reforming of glycerol, various noble-metals- and transition -metals-based catalysts on various supports with or without a promoter have been investigated. Catalyst deactivation caused by coke deposition and sintering over time is unavoidable, which consequently leads to decreases in catalytic performance and product selectivity.

ii.  Recently, intensified hybrid processes that consist of glycerol steam reforming and $CO_2$ in situ removal have been explored including sorption-enhanced steam reforming, chemical looping steam reforming, and sorption-enhanced chemical looping steam reforming. To ensure the effectiveness of these technologies, new $CO_2$ selective sorbents with better $CO_2$ sorption efficiency and simplicity in sorbent regeneration must be developed.

iii. For transformation of glycerol to fine chemicals, a wide range of catalysts based on either noble and transition metals or bimetallic systems has been developed to promote glycerol conversion and selectivity toward acrolein, lactic acid, 1,3-propanediol, or 1,2-propanediol formation. Although some previous studies demonstrated the effectiveness of some heterogeneous catalysts, the catalyst deactivation caused by coke deposition, sintering, agglomeration, and leaching remains the main technical barrier that must be addressed in future research.

iv.  To tackle this challenge, some novel reactor configurations have been designed to retard coke formation and the associated catalyst deactivation. For example, Gao et al. [38] developed a dual catalyst bed reactor for steam reforming of glycerol where $Cu/SiO_2$ is placed as the guard and $Ni/SiO_2$ is placed at the bottom to catalyze the reaction. Another instance of a new reactor design is using a membrane reactor in the steam reforming of glycerol over $Co-Ni/Al_2O_3$, as reported by Wang et al. [195]; however, it remains a necessity to design and develop efficient reactors not only for glycerol steam reforming but also for glycerol conversion to fine chemicals.

v.   Another research gap that must be filled is that, until now, most studies on glycerol transformation into value-added chemicals were performed in a batch reactor system. Even though batch reactors can effectively illustrate the operational parameters for the process, experimental data from conducting the reaction in a continuous reactor are still required for process scale-up.

**Author Contributions:** Conceptualization, Y.H. and Q.H.; methodology, Y.H.; software, Y.H.; validation, Q.H. and C.X.; formal analysis, Y.H.; investigation, Y.H. and C.X.; resources, Y.H.; data curation, Y.H.; writing—original draft preparation, Y.H.; writing—review and editing, Q.H. and C.X.; visualization, Y.H.; supervision, Q.H. and C.X.; project administration, Q.H. and C.X.; funding acquisition, Q.H. and C.X. All authors have read and agreed to the published version of the manuscript.

**Funding:** This research received no external funding.

**Data Availability Statement:** Data is contained within the article.

**Acknowledgments:** The authors would like to acknowledge funding from the Natural Sciences and Engineering Research Council of Canada (NSERC) for Discovery Grants, and the Startup Fund from the University of Prince Edward Island.

**Conflicts of Interest:** The authors declared they have no conflict of interest.

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
