# Peer review of "Catalytic Conversion of Glycerol into Hydrogen and Value-Added Chemicals: Recent Research Advances"

_catalysts, doi:10.3390/catal11121455_

Round 1
Reviewer 1 Report
I have reviewed the manuscript in detail, and I consider it to be a good work; however, it is important to improve some points that I mention below:
Line 80-82. Delete: including steam reforming to produce H2, de- 80 hydration to acrolein, oxidation to lactic acid, hydrogenolysis to 1,3-propanediol and 1,2- 81 propanediol.. it is repeated information (presented in lines 62-63).
Lines 724-728. Mention this information in the body of the text and not in the conclusions.
Improve the quality of the figures. They look very blurry, mainly figures 1-3 and 8-12.
Author Response
Reviewer-1:
I have reviewed the manuscript in detail, and I consider it to be a good work; however, it is important to improve some points that I mention below:
[1] Line 80-82. Delete: including steam reforming to produce H2, de- 80 hydration to acrolein, oxidation to lactic acid, hydrogenolysis to 1,3-propanediol and 1,2- 81 propanediol.. it is repeated information (presented in lines 62-63).
Response: Thank you for the comment. The repeated sentence has been removed from the Page 4 of the revised manuscript.
[2] Lines 724-728. Mention this information in the body of the text and not in the conclusions.
Response: As kindly suggested by the reviewer, the sentences have been added in the Page 13 of the revision and some changes have been made as well, “Until now, although steam reforming is the most dominant conversion route to produce H2, several emerging H2 production technologies have also been developed for glycerol such as photo-reforming and catalytic transfer hydrogenation, which still require more studies to illustrate the underlying mechanism, develop more efficient catalysts, and further optimize the operating parameters for better conversion efficiency and selectivity [5]”.
[3] Improve the quality of the figures. They look very blurry, mainly figures 1-3 and 8-12.
Response: In accordance with the reviewer’s comment, the quality of figures has been improved in the revised manuscript.
Reviewer 2 Report
In this review paper, you investigated"Catalytic Conversion of Glycerol into Hydrogen and Value-added Chemicals: Recent Research Advances".A good review paper should provide readers with complete information about the topic. It is better to add the following content to the manuscript:
1-One of the uses glycerol can be used to produce biodiesel, so it is better to refer to biodiesel production methods, production efficiency, catalysts, and quality.
2-It is best to dedicate a section to how the catalyst is made, the manufacturing conditions, and the tests needed to ensure the catalyst specifications.
3-One of the important issues in the use of catalysts is the selection of optimal conditions that can be achieved with experimental design methods. So this section should be added to the article.
4-Some of the findings and the importance of this manuscript should be mentioned in detail in the light sections.
Author Response
Reviewer-2:
In this review paper, you investigated "Catalytic Conversion of Glycerol into Hydrogen and Value-added Chemicals: Recent Research Advances". A good review paper should provide readers with complete information about the topic. It is better to add the following content to the manuscript:
[1] One of the uses glycerol can be used to produce biodiesel, so it is better to refer to biodiesel production methods, production efficiency, catalysts, and quality.
Response: The authors totally agree with the reviewer that the biodiesel production approaches from glycerol and the associated production efficiency, catalysts, and quality should be included to make this review article more comprehensive. New sentences have been added in the Page 2 of the revised manuscript, “Aside from H2 and chemicals, glycerol can also be applied as the feed to react with free fatty acids to form glycerides by glycerolysis (or called glycerol esterification), and the resulting glyceride can be further treated to produce biodiesel via alkaline transesterification. Nevertheless, owing to the use of high-cost metallic catalysts and higher temperature (up to ~ 200 ℃), glycerolysis is not a common technology used in the biodiesel industry rather than being widely employed in the cosmetic, pharmaceutical, and food industries to synthesize surfactants and emulsifiers [191]. The associated underlying mechanism, major reaction conditions (temperature, reactor configuration, molar ratio of glycerol and free fatty acid, type of free fatty acid, catalyst type, and glycerol purity), and technical challenges have been recently reviewed by Mamtani et al. [192] and Abomohra et al. [193]”.
[2] It is best to dedicate a section to how the catalyst is made, the manufacturing conditions, and the tests needed to ensure the catalyst specifications.
Response: Thank you for this comment. A new section has been added in the Page 25-26 of the revision, as shown below:
“4. General catalyst preparation strategies and associated characterization methods
In addition to the type of active metal and catalyst support, catalyst preparation is considered as another important factor affecting the catalytic performance and product yield and selectivity in the catalytic transformation of glycerol to fuels and chemicals [194]. The most commonly used catalyst preparation methods in the catalytic glycerol valorization include: impregnation [195-196], hydrothermal method [197], precipitation [198], sol-gel [199], and wet incipient [200]. Among them, the commonly used methods for catalyst preparation are impregnation and precipitation. The impregnation is mainly dependent on the interaction between the surface of the support and the species in the solution prepared, which can be further divided into wet impregnation and dry impregnation according to the volume of impregnation solution introduced to the pores of the support material. In comparison, the main benefit offered by dry impregnation is that the amount of the added components to the catalyst is easier to be controlled than wet impregnation; however, the synthesized catalyst by dry impregnation might not be as uniform as the catalyst prepared by wet impregnation [201]. On the other hand, precipitation is the most widely applied catalyst preparation method because of its low-cost and facile. For example, there are several industrially used catalysts are prepared by precipitation or co-precipitation such as SiO2-Al2O3 used in the Fluid Catalytic Cracking (FCC), Fe2O3 applied in the Fisher Tropsch process, and Cu-ZnO/Al2O3 employed in the methanol synthesis [202]. To the best of our knowledge, there is no study till now has compared different catalysts preparation methods in terms of their activity, stability, and reducibility during the glycerol transformation, which could be one of the future directions. After the catalyst synthesis, a series of analytical techniques are needed to determine the characteristics of the prepared catalysts, and the most widely applied characterization methods like X-ray fluorescence spectrometer, N2 adsorption-desorption analyzer, X-ray diffractometer (XRD), transmission electron microscopy (TEM), and temperature-programmed desorption of ammonia (NH3-TPD) analyzer [145] [203]”.
[3] One of the important issues in the use of catalysts is the selection of optimal conditions that can be achieved with experimental design methods. So this section should be added to the article.
Response: As suggested by the reviewer, the following sentences have been added in the Page 13 of the revision, “In this section, the catalytic transformation of glycerol as the feedstock to produce acrolein by dehydration, lactic acid by oxidation, and 1,3-propanediol and 1,2-propanediol via selective hydrogenolysis is discussed, with a focus on the effect of the type of active metal specie and catalyst support on the product yield and selectivity and catalyst deactivation. It is undoubtedly that the reaction conditions play an important role in regulating the reaction and affecting the catalytic performance. Previously, the influence of operating conditions on the glycerol conversion to acrolein, lactic acid, and propanediols has been thoroughly reviewed by Belousov [54], Abdullah et al. [204], and Vasiliadou and Lemonidou [205], respectively”.
[4] Some of the findings and the importance of this manuscript should be mentioned in detail in the light sections.
Response: Thank you for this comment. The following sentences have been added in the revised manuscript to highlight the most important findings/conclusions from each section.
Page 12-13: “To date, a wide range of noble metals- and transition metals-based catalysts supported on various supports with or without a promoter have been heavily tested to produce H2 from glycerol by steam reforming; however, the catalyst deactivation caused by coke deposition and sintering over time is unavoidable and consequently leads to a decrease in the catalytic performance and product selectivity. The detailed catalyst deactivation mechanism during the glycerol steam reforming has been previously reviewed by Roslan et al. [5]. In particular, the cost for replacing fresh catalyst and shutdown the industrial processes could be billions of dollars in general [5]. Despite the fact that poisoning is another cause for the loss in the catalytic activity, the compounds that could lead to the poisoning in the steam reforming are typically absent and thus more efforts must be focused on the coke deposition and sintering of metal particles [14]. On the other hand, Lehnert and Claus [39] reported that the presence of NaCl in the crude glycerol led to the poisoning of metal species, thereby causing a low crude glycerol conversion and fast catalyst deactivation. In addition to the new catalysts, a great deal of effort has also been applied in the development of novel reactor configurations to further enhance production efficiency include SESR, CLSR, and SECLR. For instance, SESR is capable for achieving in-situ CO2 removal and thus shifts the water gas shift reaction (Eqn. 3) towards producing more H2 gas and simultaneously limits the methanation (Eqn. 5) and coke formation (Eqn. 12). In short, these newly developed technologies provide benefits to glycerol steam reforming by retarding the side reactions and hence promoting H2 formation [14].
Until now, although steam reforming is the most dominant conversion route to produce H2, several emerging H2 production technologies have also been developed for glycerol such as photo-reforming and catalytic transfer hydrogenation, which still require more studies to illustrate the underlying mechanism, develop more efficient catalysts, and further optimize the operating parameters for better conversion efficiency and selectivity [5]”.
Page 25: “Overall, the transformation of glycerol to value-added C3 chemicals (i.e., acrolein, lactic acid, 1,3-propanediol, and 1,2-propanediol) over heterogeneous catalysts is an essential component for achieving sustainable and economic production of biodiesel. To ensure a high yield of target products, the proper catalyst design including the selection of active metal specie, support material, catalyst preparation method, and the type and strength of acid sites) and reaction conditions must be conscientiously determined [206]”.